# GS$^2$-GNeSF: Geometry-Semantics Synergy for Generalizable Neural Semantic Fields

## ABSTRACT

The remarkable success of neural radiance fields in low-level vision tasks such as novel view synthesis has motivated its extension to high-level semantic understanding, giving rise to the concept of the neural semantic field (NeSF). NeSF aims to simultaneously synthesize novel view images and associated semantic segmentation maps. Generalizable NeSF, in particular, is an appealing direction as it can generalize to unseen scenes for synthesizing images and semantic maps for novel views, thereby avoiding the need for tedious per-scene optimization. However, existing approaches to generalizable NeSF fall short in fully exploiting the geometric and semantic features as well as their mutual interactions, resulting in suboptimal performance in both novel-view image synthesis and semantic segmentation. To address this limitation, we propose Geometry-Semantics Synergy for Generalized Neural Semantic Fields (GS$^2$-GNeSF), a novel approach aimed at improving the performance of generalizable NeSF through the comprehensive construction and synergistic interaction of geometric and semantic features. In GS$^2$-GNeSF, we introduce a robust geometric prior generator to generate the cost volumes and depth prior, which aid in constructing geometric features and facilitating geometric-aware sampling. Leveraging the depth prior, we additionally construct a global semantic context for the target view. This context provides two types of compensation information to enhance geometry and semantic features, achieved through boundary detection and semantic segmentation, respectively. Lastly, we present an efficient dual-directional interactive attention mechanism to foster deep interactions between the enhanced geometric and semantic features. Experiments conducted on both synthetic and real datasets demonstrate that our GS$^2$-GNeSF outperforms existing methods in both novel view and semantic map synthesis, highlighting its effectiveness in generalizing neural semantic fields for unseen scenes.

## KEYWORDS

Generalizable Neural Radiance Fields, Generalizable Neural Semantic Fields, Novel View Synthesis, Semantic Segmentation

## 1 INTRODUCTION

Recently, Neural Radiance Field (NeRF) [22] has emerged as a focal point in the field of computer vision due to its outstanding performance in novel view synthesis. To extend the capability of NeRF

*ACM MM, 2024, Melbourne, Australia*

© 2024 Copyright held by the owner/author(s). Publication rights licensed to ACM.
ACM ISBN 978-x-xxxx-xxxx-x/YY/MM
https://doi.org/10.1145/nnnnnnn.nnnnnnn

in understanding scene semantics, researchers have explored the Neural Semantic Field (NeSF) [2, 10, 15, 18, 31, 40], which synthesizes associated semantic segmentation maps alongside novel view images by mapping spatial coordinates to semantic labels. This advancement enables a deep comprehension of scene semantics, facilitating NeSF's application in tasks requiring detailed scene understanding and segmentation, such as autonomous driving, augmented reality, and robotic navigation.

While NeSF presents an attractive direction, many existing NeSF methods require a labor-intensive optimization process on a per-scene basis, as they are typically built on top of NeRF. To address this challenge and inspired by the idea of generalizable NeRF [14, 21, 33], several studies [4, 7, 20] have shifted their focus towards generalizable NeSF learning. In this framework, the goal is to synthesize novel view images and semantic maps of unknown scenes from arbitrary views without per-scene optimization. To achieve this goal, S-Ray [20] decouples the tasks of image synthesis and semantic prediction, building upon NeuRay [21] to address the former task while leveraging ray-level semantic context to predict semantics. GSNeRF [7] also employs a decoupled framework but focuses on extracting explicit geometric features for image synthesis. It additionally estimates depth maps of the target view using geometric information for efficient sampling, thus avoiding the expensive hierarchical sampling strategy used in S-Ray. In contrast to decoupling the two branches, GNeSF [4] derives geometric features from an explicit feature volume initialized via Multi-View Stereo (MVS) and concatenates these geometric features with semantic features to make semantic predictions. Despite their endeavors, these generalizable NeSF approaches fail to fully exploit robust geometric and semantic features, as well as their mutual interactions. These limitations lead to suboptimal performance in both novel-view image synthesis and semantic segmentation, as shown in Table 1.

Inspired by insights gleaned from recent studies [5, 12, 13, 17, 27, 39], we believe that the geometric understanding (*e.g.* volume density) and semantic understanding (*e.g.* semantic labels) carry mutually beneficial information: geometric understanding can offer essential clues such as object shape and depth to aid in segmentation, while semantic understanding provides semantic context such as class relationships to support geometry prediction. To this end, we introduce a novel Geometry-Semantics Synergy method for Generalized Neural Semantic Fields, named GS$^2$-GNeSF, which aims to improve the performance of both novel-view image synthesis and semantic segmentation for novel views in unseen scenes by leveraging the synergy between geometry and semantics.

In our GS$^2$-GNeSF, we introduce a robust geometric prior generation (RGPG) module to generate cost volumes and depth prior (*i.e.*, depth map and depth hypotheses) for the target view. These priors serve as robust geometric representations and guide our sampling process. By employing geometry-aware sampling based on the depth prior, we can efficiently generate our initial geometric and semantic features. Additionally, with the predicted depth

map, we establish a global semantic context for the target view to model inter-ray context on a global scale. We develop a global context compensation module, which includes a dual-branch encoder network, to process this context and generate compensation information for enhancing semantic and geometric features. Specifically, we introduce semantic segmentation and boundary detection as two auxiliary tasks within this encoder. These tasks guide the generation of class-specific and boundary-aware features to serve as compensation. Lastly, we present an efficient dual-directional interactive attention mechanism to facilitate the deep interactive fusion of the enhanced geometric and semantic features from different perspectives, *i.e.*, inter-view and intra-view.

Our geometry-semantics synergy is manifested in three aspects: 1) *geometric and semantic features construction* (Sec. 3.1): We construct initial geometric and semantic features through geometry-aware sampling, which is realized by depth prior generated by our designed robust geometric prior generation module. 2) *global context compensation* (Sec. 3.2): We enhance the geometric and semantic features by leveraging two types of compensation information learned from our global semantic context, which is constructed through semantic information in conjunction with depth prior. 3) *GeoSem interaction* (Sec. 3.3): We achieve a deep interactive fusion between semantic and geometric information via our efficient dual-directional interactive attention mechanism.

Through extensive experiments conducted on both synthetic and real datasets, we demonstrate that our geometry-semantics synergy method outperforms existing generalizable NeSF methods in both image and semantic map synthesis for unseen scenes under either generalization or fine-tuning settings. By ablating each module in our GS$^2$-GNeSF, we showcase their individual effects in fostering synergy between geometry understanding and semantic understanding.

## 2 RELATED WORK

**Neural Radiance Fields.** Neural Radiance Fields (NeRF) [22] represent a novel technique to novel view synthesis based on implicit neural representations, employing Multilayer Perceptrons (MLPs) to learn continuous 3D scenes. By leveraging multiple images from different viewpoints, NeRF can render unseen viewpoints with remarkable detail fidelity. This technique has demonstrated exceptional capabilities in synthesizing new viewpoints, yielding impressive demonstrations that have catalyzed a wave of subsequent research efforts. Despite its strengths, NeRF is significantly hampered by its reliance on computationally intensive scene-specific optimization processes.

To address this limitation, the concept of generalizable neural radiance fields [3, 14, 33, 36, 38] has emerged, focusing on generalization across different scenes by learning representations of radiance fields directly from a given set of scene images. Previous methods leverage either warped 2D [33, 36, 38] or 3D [3, 14] features from nearby reference views as network inputs to condition NeRF rendering, instead of directly inputting positional encoding [22]. As a result, they can synthesize novel view images of unseen scenes without per-scene optimization.

**NeRF with Depth Priors.** Several research efforts [9, 24, 26, 32, 34] have explored the incorporation of depth priors to enhance the

capabilities of NeRF. These endeavors utilize techniques like multi-view stereo or monocular depth estimation to produce depth priors, which are subsequently integrated into NeRF to guide the sampling process and refine the rendering of 3D scenes. [9, 24, 34] have shown that leveraging depth priors can significantly improve both the training efficiency and the inference performance of NeRF, leading to notable reductions in the number of sampling points required and the overall training duration.

Furthermore, recent works [14, 21, 26, 32] integrate depth priors into generalizable NeRF models for novel view synthesis. These methods leverage extracted depth information to navigate the sampling procedure [26, 32] or to act as a criterion for evaluating for the visibility of reference views [14, 21]. NeRF-SDP [32] and Garf [26], which regress coarse depth maps on target views to guide sampling, share similarities with our approach. However, unlike their method of constructing the cost volume, which relies on 2D feature maps from nearby reference views to construct the cost volume for the target view, we employ 3D features extracted from these reference views. This choice is motivated by the fact that, in contrast to 2D features, 3D features derived from cost volumes encapsulate more comprehensive and robust information.

**Neural Semantic Fields.** Neural semantic field (NeSF) [2, 4, 7, 10, 15, 18, 20, 31, 40] extends the NeRF framework to output additional semantic labels by incorporating semantics into the neural representation. Semantic-NeRF [40], as a pioneering work in this domain, integrates semantic labeling directly into the NeRF output through the augmentation the vanilla NeRF architecture with an additional branch dedicated to semantic prediction. Several subsequent research endeavors [2, 10, 15, 18] have built upon Semantic-NeRF, aiming to embed semantic features to enhance its capabilities. For instance, in [2, 15], the semantic features from CLIP [25] or SAM [16] are explored and integrated into Semantic-NeRF architecture, enriching the semantic understanding and interpretability of the generated scenes.

Generalizable NeSF approaches [4, 7, 20], bypass the need for per-scene optimization, offering effective generalization to unseen scenes. S-Ray [20] separates novel-view image synthesis and semantic segmentation into two branches, utilizing IBRNet for image synthesis and a novel cross-reprojection attention for semantic context. However, it sometimes produces artifacts due to limited geometric representation. To address this limitation, GSNeRF [7] proposes a semantic geo-reasoning module to extract additional 3D volume features and estimate depth maps from reference images. Leveraging estimated depth information, GSNeRF samples points along rays more effectively and employs 3D volume features as additional conditions for volume rendering. Similarly, GNeSF [4] constructs an explicit feature volume via MVS to serve as geometric features. These features not only contribute to volume rendering but also serve as inputs, concatenated with multi-view image features, to generate soft voting weights for aggregating semantic maps predicted for reference views.

## 3 METHOD

In the task of generalizable neural semantic fields, for any *unseen* scene, provided with the camera pose of a target view and a set of nearby reference views' RGB images and corresponding poses (*i.e.*,

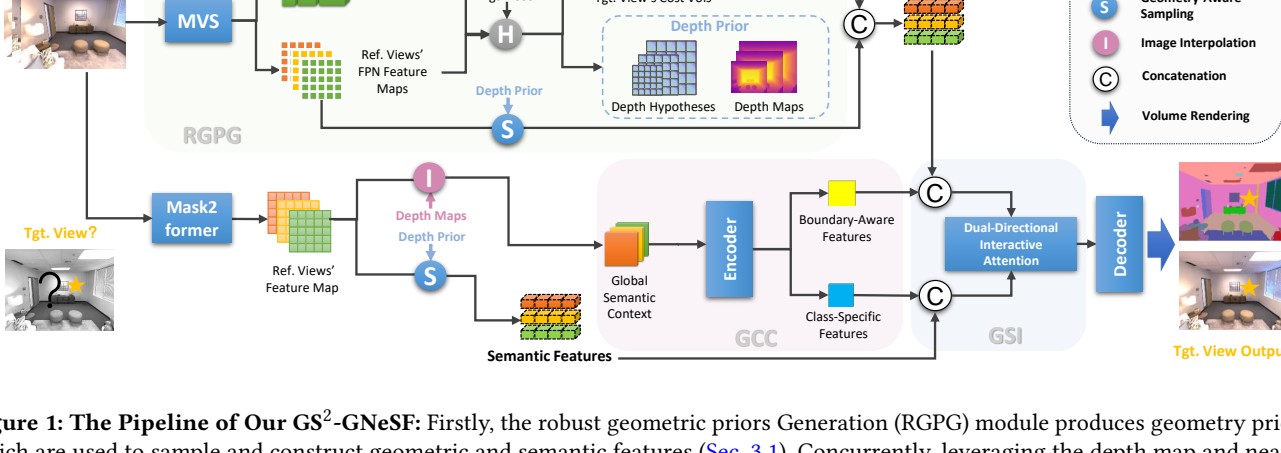

**Figure 1: The Pipeline of Our GS²-GNeSF:** Firstly, the robust geometric priors Generation (RGPG) module produces geometry priors, which are used to sample and construct geometric and semantic features (Sec. 3.1). Concurrently, leveraging the depth map and nearby reference views' semantic maps, the Global Context Compensation (GCC) module produces boundary-aware and class-specific features (Sec. 3.2). These compensation features as well as geometric and semantic features are fed into the GeoSem Interaction (GSI) module outputting deeply fused geometric and semantic features (Sec. 3.3). Lastly, the fused geometric and semantic features are input into an MLP-based decoder and respectively render final predicted color image and semantic map for the target view via volume rendering (Sec. 3.4).

posed images), our goal is to synthesize the RGB image and semantic map for this novel (target) view. To achieve this goal, we propose a novel **G**eometry-**S**emantics **S**ynergy method for **G**eneralizable **Ne**ural **S**emantic **F**ields, named GS²-GNeSF.

The framework of our GS²-GNeSF is illustrated in Figure 1, which includes three key components: 1) a **Robust Geometric Priors Generation** (RGPG) module, providing target view's geometry priors that consist of a robust 3D geometric representation in the form of cost volume and depth prior represented by the depth map and hypotheses. 2) a **Global Context Compensation** (GCC) module, offering semantic priors in a global context to enhance the geometric and semantic features through compensation supervised by two types of auxiliary targets, *i.e.*, boundary detection and semantic segmentation. 3) a **GeoSem Interaction** (GSI) module, fusing the enhanced geometry and semantic features deeply and interactively through dual-directional interactive attention.

## 3.1 Geometric & Semantic Feature Construction

*3.1.1* ***Robust Geometric Prior Generation (RGPG)***. We design RGPG to provide our model with two types of geometry priors for the target view: 1) *cost volume*, which serves as a robust geometric representation and is widely used in depth prediction tasks [11, 37]. 2) *depth prior* (depth map and depth hypotheses), which enables geometry-aware sampling on a ray and facilitates more accurate and efficient sampling.

When employing traditional multi-view stereo (MVS) methods [11, 37] to estimate the cost volume and depth map for a view, the color image of that view is typically required. However, in the context of novel view synthesis, the color image of the target view is unavailable. To address this limitation, we extract cost volumes from nearby reference views and utilize the camera pose of the target view to construct the geometry priors cascadingly. As illustrated

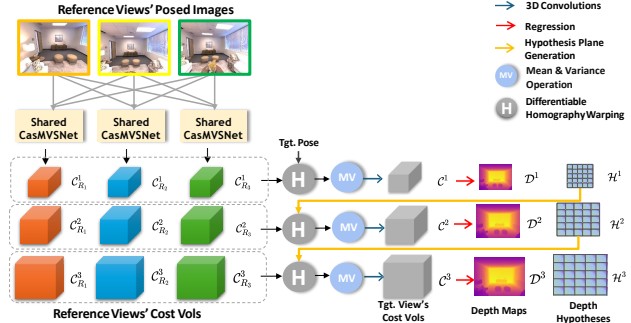

**Figure 2: Robust Geometric Prior Generation.** We extract information from nearby $V$ reference views to construct multi-level cost volumes, applying differentiable homography warping to produce the target view's depth, depth hypotheses, and cost volumes for the subsequent synergy between geometry and semantics.

in Figure 2, we adopt CasMVSNet [11] to construct multi-level cost volumes $\{C_{R_i}^l\}_{l=1}^L$ for each reference view $R_i$, using the remaining $(V-1)$ reference views. We set $L$ to 3, with each level representing different degrees of granularity. $C_{R_i}^l \in \mathbb{R}^{F_l \times S_l \times H_l \times W_l}$ denotes the $l$-th cost volume for the reference view $R_i$, where $F_l$ is the number of channels, $S_l$ indicates the number of depth hypotheses, and $H_l \times W_l$ represents the resolution at this level. Subsequently, we create the multi-level cost volumes $\{C^l\}_{l=1}^L$ for the target view cascadingly in a coarse-to-fine manner akin to [11], by using the depth hypotheses[1] from the previous level. Specifically, we first apply differential homography [37] to warp the cost volume of $V$ reference views at $l$-th level $\{C_{R_v}^l\}_{v=1}^V$, resulting in $V$ warped cost volumes. The

---
[1]When $l = 1$, we pre-define a depth range as depth hypotheses.

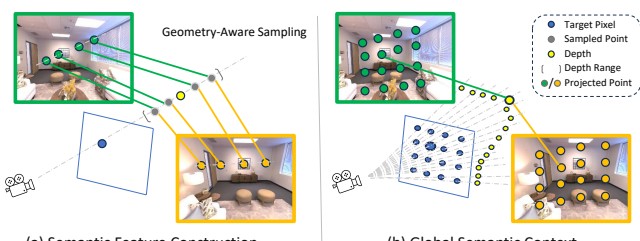

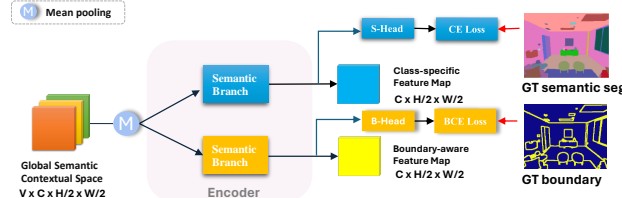

Figure 3: (a) Semantic feature construction involves geometry-aware sampling and the gathering of multi-view features obtained by projecting sampled points onto reference views. (b) Global semantic context is efficiently constructed for the target view via using grid sampling and the estimated depth map.

mean and variance of these volumes are then concatenated and passed through a 3D convolutional neural network to generate the cost volume for the target view $C^l$. Finally, a regression head is applied to $C^l$ to produce a depth map $\mathcal{D}^l \in \mathbb{R}^{H_l \times W_l}$ and depth hypotheses $\mathcal{H}^l \in \mathbb{R}^{S_l \times H_l \times W_l}$. Consequently, we obtain multi-level cost volumes $\{C^l\}_{l=1}^L$, depth maps $\{\mathcal{D}^l\}_{l=1}^L$, and depth hypotheses $\{\mathcal{H}^l\}_{l=1}^L$ for the target view, serving as our geometry priors.

*3.1.2  **Geometry-Aware Sampling**.* Sampling plays a crucial role in neural rendering [7, 32], influencing both rendering quality and computational efficiency. Aggregating features from noisy points along the entire ray can lead to subpar rendering results. Moreover, traditional sampling methods in NeRF often necessitate sampling a large number of points along a ray [4, 20, 22], which is time-consuming and computationally expensive. To address these challenges and achieve more accurate and efficient sampling, we introduce a *geometry-aware sampling* strategy utilizing our generated depth prior. Upon obtaining the depth maps and depth hypotheses for the target view, as shown in Figure 3 (a), we can apply our geometry-aware sampling strategy to skip empty space in the scene. Specifically, for each ray emitted from a pixel, we uniformly sample within the range of the depth hypotheses at the finest level, *i.e.*, $\mathcal{H}^L$, to acquire $M$ sample points. Each pixel's depth range spans from the nearest to the farthest plausible depths. In this paper, we only sample *8 points* for each ray, which is significantly fewer than 128, the number of points sampled by S-Ray [20] and GNeSF [4].

*3.1.3  **Geometric Features Construction**.* For the $M$ sampled points along a ray, we construct their geometric features by incorporating both view-dependent and view-independent information, representing geometric details from both reference views and the target view. The view-dependent features $\mathbf{T}_{\text{dep}} \in \mathbb{R}^{M \times V \times F_0}$ are derived by interpolating intermediate feature maps of $V$ reference images from the final output of the Feature Pyramid Network (FPN) [19] in CasMVSNet. Conversely, the view-independent features $\mathbf{T}_{\text{indep}} \in \mathbb{R}^{M \times (F_1+F_2+F_3)}$ are obtained by interpolating cost volume at each level $C^l$ for the corresponding $M$ points, followed by concatenating over three levels. After obtaining $\mathbf{T}_{\text{dep}}$ and $\mathbf{T}_{\text{indep}}$, they are concatenated[2] and passed through a Multi-Layer Perceptron (MLP) to generate our geometric feature $\mathbf{G} \in \mathbb{R}^{M \times V \times C}$.

---

[2]To concatenate, $\mathbf{T}_{\text{indep}}$ is tiled for $V$ times to match the shape of $\mathbf{T}_{\text{dep}}$.

Figure 4: Global Context Compensation. The global context compensation employs a dual-branch 2D model on global semantic context, with each branch equipped with a dedicated output head for its respective task, generating corresponding feature maps tailored to those tasks.

*3.1.4  **Semantic Features Construction**.* We construct our semantic features $\mathbf{S} \in \mathbb{R}^{M \times V \times C}$ similarly to S-Ray [20]. Firstly, we use a sophisticated feature extractor, *i.e.*, Mask2Former [6], to obtain $C$-dimensional feature maps $\mathbf{F}_i \in \mathbb{R}^{C \times H \times W}$ for each reference view. Unlike S-Ray, we employ geometry-aware sampling for each target pixel, allowing us to sample points closer to object surfaces. This results in a more compact sampling set compared to S-Ray, with fewer noise features introduced. Subsequently, our sampled points $\{p_j\}_{j=1}^M$ are projected onto the reference views using the camera calibration matrix $K$, rotation matrix $R$, and translation vector $t$, yielding $p_j^* = K \cdot \pi \cdot (R \cdot p_j + t)$, where $\pi$ denotes the projection function. Finally, the features at the projected points $\{\mathbf{F}_i(p_j^*)\}$ are gathered as our semantic features $\mathbf{S}$.

## 3.2  Global Context Compensation (GCC)

Based on predicted depth map of the target view $\mathcal{D}^L$ and semantic information extracted from reference views $\{\mathbf{F}_i\}_{i=1}^V$, we design a Global Context Compensation (GCC) module to compensate both semantic and geometric features via a global semantic context. This context serves as a semantic prior capturing essential inter-ray relationships within the target view, as illustrated in Figure 3 (b). Employing this global semantic context, we utilize a dedicated encoder network designed to generate two types of compensation information: one aims at enhancing geometric features and the other at enriching semantic features. This is achieved through the utilization of two auxiliary tasks: boundary detection and semantic segmentation, respectively, as illustrated in Figure 4.

**Global Semantic Context Construction.** Unlike the semantic features constructed in Sec. 3.1.4, which gather information from reference views for a single ray, our global semantic context aims to encompass all rays within the target view. As such, it can serve as a semantic prior, providing valuable information for the entire target view. To efficiently construct the global semantic context while minimizing computational costs, we utilize a combination of grid sampling and our generated depth prior. As illustrated in Figure 3 (b), we perform grid sampling on the target view, yielding rays at a resolution of $H/2 \times W/2$. We then sample a single point for each ray based on the depth map and collect the corresponding features from the reference views using point projections. This process allows us to construct a global semantic context denoted as $\mathbf{A} \in \mathbb{R}^{(H/2 \times W/2) \times V \times C}$.

**Figure 5: GeoSem Interaction.** We first construct semantic and geometric inputs. Subsequently, the Dual-Directional Interactive Attention deeply integrates semantic and geometric information through sequential inter-view and intra-view interactions.

**Compensation Information Generation.** We believe that this global semantic context, serving as a semantic prior, has the potential to offer not only the semantic context of the target view but also to capture critical high-frequency information at boundaries within the target view. The incorporation of this boundary-aware information could enhance our geometric features, which can be considered as a synergistic interplay between geometry and semantics. As a result, we introduce the GCC module, as illustrated in Figure 4, which comprises a dual-branch encoder and two auxiliary headers. The encoder's semantic branch is dedicated to capturing high-level semantics and producing class-specific features. On the other hand, the boundary branch of the encoder focuses on capturing high-frequency information at class boundaries and generating boundary-aware features. By adding an auxiliary header for each branch with the corresponding supervision, the GCC module can effectively generate compensation information in terms of semantic context and boundary details.

Specifically, given the global semantic context $\mathbf{A}$, we first perform mean pooling over $V$ views, resulting in $\mathbf{A}' \in \mathbb{R}^{H/2 \times W/2 \times C}$. $\mathbf{A}'$ is then fed into both the semantic and boundary branch of the encoder, yielding class-specific features $\mathbf{F}_{cls} \in \mathbb{R}^{H/2 \times W/2 \times C}$ and boundary-aware features $\mathbf{F}_{bon} \in \mathbb{R}^{H/2 \times W/2 \times C}$, respectively. A semantic head (S-Head) is applied to the semantic branch to predict semantic labels of the target view, which are supervised using a cross entropy loss (CE-Loss). Simultaneously, a boundary head (B-Head) is employed to the boundary branch to perform boundary detection, whose predictions are supervised using a boundary-awareness cross entropy loss [35] (BCE-Loss). A detailed description of the encoder network architecture is provided in the supplementary material.

## 3.3 GeoSem Interaction (GSI)

We introduce a GeoSem Interaction (GSI) module to enhance and deeply fuse the geometric and semantic features, thereby improving the model's ability to reconstruct and understand complex unseen scenes. The details of our GSI module is illustrated in Figure 5. Within our GSI module, we first enhance the geometric features $\mathbf{G}$ and semantic features $\mathbf{S}$ using compensation information generated

by the GCC module, as expressed by:

$$\mathbf{G}' = [\mathbf{G}; \mathbf{F}_{bon}]; \quad \mathbf{S}' = [\mathbf{S}; \mathbf{F}_{cls}], \tag{1}$$

where $[\star; \star]$ denotes concatenation after shape broadcasting. Subsequently, a dual-directional interactive attention mechanism is designed to process $\mathbf{G}'$ and $\mathbf{S}'$ and sequentially perform interactive deep fusion of semantic and geometric information from inter-view and intra-view perspectives.

**Dual-Directional Interactive Attention.** A straightforward way to fuse $\mathbf{G}'$ and $\mathbf{S}'$ involves directly applying cross attention to them. However, this method of fusion can be computational expensive, since each ray's feature is a rank-3 tensor (*e.g.*, $\mathbf{G}'$ or $\mathbf{S}'$ in Figure 5). Inspired by S-Ray [20], we design a dual-directional interactive attention mechanism that applies multi-head cross-attention along $V$ and $M$ dimension sequentially. Dual-directional interactive attention not only maintains the lightweight computational and memory efficiencies characteristic of S-Ray's design but also facilitates the integration of geometric and semantic features, a capability not present in S-Ray.

Specifically, our dual-directional interactive attention consists of two consecutive phases: inter-view phase and intra-view phase, as shown in Figure 5. In inter-view phase, the input features with the shape $[M, V, 2C]$ is reorganized to a set of $V \times 2C$ slice with batch size $= M$. Each slice corresponds to the multi-view features of specific point. For instance, $\mathbf{S}'$ is reorganized as $\{\mathbf{S}'_i\}_{i=1}^{M}$, wehre $\mathbf{S}'_i$ denotes $i$-th slice of $\mathbf{S}'$. Subsequently, a multi-head cross attention [30] block is applied on slices to perform 1D attention with one type of features as query and the other type of features as key and value. Taking slice $\mathbf{S}'_i$ as query and $\mathbf{G}'_i$ as key and value. Formally, the multi-head cross attention block's output $\mathbf{S}''_i$ is formulated as:

$$Q^{(h)} = \mathbf{S}'_i W_q^{(h)}, \quad K^{(h)} = \mathbf{G}'_i W_k^{(h)}, \quad V^{(h)} = \mathbf{G}'_i W_v^{(h)}, \tag{2}$$

$$A^{(h)} = \sigma\left(\frac{Q^{(h)} K^{(h)T}}{\sqrt{d_k}}\right) V^{(h)}, \quad h = 1, \dots, H, \tag{3}$$

$$x = [A^{(1)}; \dots; A^{(H)}] W_o, \tag{4}$$

$$y = LayerNorm(\mathbf{S}'_i + x), \tag{5}$$

$$\mathbf{S}''_i = LayerNorm(y + FFN(y)), \tag{6}$$

| Method | Settings | Synthetic Data (Replica) | | | | | Real Data (ScanNet) | | | | |
|--------|----------|--------|--------|--------|-----------|--------|--------|--------|--------|-----------|--------|
| | | PSNR↑ | SSIM↑ | mIoU↑ | Total Acc↑ | mAcc↑ | PSNR↑ | SSIM↑ | mIoU↑ | Total Acc↑ | mAcc↑ |
| S-Ray [20] | Generalization | 30.12 | 0.932 | 41.59 | 70.51 | 47.19 | 26.57 | 0.832 | 57.15 | 78.24 | 62.55 |
| GSNeRF [7] | | 31.16 | 0.924 | 51.52 | 83.41 | 61.29 | 31.33 | 0.907 | 58.30 | 79.79 | 65.93 |
| GNeSF [4] | | 31.90 | 0.948 | 55.21 | 77.64 | 61.48 | 24.44 | 0.858 | 71.60 | 87.60 | 82.30 |
| **Ours** | | **34.49** | **0.962** | **61.75** | **80.45** | **66.30** | **32.22** | **0.931** | **74.97** | **90.40** | **83.64** |
| S-Ray ft [20] | Finetuning | 32.21 | 0.966 | 75.96 | 96.38 | 80.81 | 29.27 | 0.865 | 91.06 | 98.20 | 93.97 |
| GSNeRF ft [7] | | - | - | - | - | - | 31.70 | - | 93.20 | 99.10 | 98.40 |
| GNeSF ft [4] | | 32.13 | 0.963 | 76.71 | 96.59 | 83.57 | 29.22 | 0.871 | 93.32 | 96.33 | 98.51 |
| **Ours ft** | | **34.56** | **0.974** | **87.17** | **97.77** | **86.20** | **34.13** | **0.962** | **93.33** | **99.21** | **98.53** |

**Table 1:** Comparisons with baseline methods for generalizable NeSF under both *generalization* and *fine-tuning* settings.

where H is the number of heads, $\sigma(\cdot)$ denotes the softmax function, and $d_k = C/H$ is the dimension of each head. The term $A^{(h)}$ indicates the output from the $h$-th attention head, $Q^{(h)}, K^{(h)}, V^{(h)} \in \mathbb{R}^{d_k}$ denote query, key, and value correspondent to $h$-th attention head. $W_q^{(h)}, W_k^{(h)}, W_v^{(h)} \in \mathbb{R}^{C \times d_k}$, and $W_o^{(h)} \in \mathbb{R}^{C \times C}$ are the projection matrices. FFN stands for feed-forward network [30], Layer-Norm denotes layer normalization [1]. With this process, semantic and geometric features exchange information from the *inter-view* perspective. Stepping into *intra-view* phase, the output feature of previous phase is reorganized to a set of $M \times 2C$ slice with batch size = V, where each slice represents target ray's features correspondent to specific reference view. After applying multi-head cross-attention over semantic and geometric features, the intra-view semantic and geometric information can be fused within each view, outputting final semantic features $S^*$ and geometric features $G^*$.

## 3.4 Final Predictions and Optimization

**Final Predictions.** Both $G^*$ and $S^*$ are fed into an MLP-based decoder for subsequent predictions. Within the decoder, we first predict a view weighting vector $w$ based on $G^*$, following IBR-Net [33]. This $w$ indicates the importance of reference views and is subsequently employed for weighted pooling on $S^*$ and $G^*$ to reduce the $V$ dimension. Following this, we predict the semantic labels and density estimates of sampled points along a ray using the pooled $S^*$ and $G^*$. Additionally, the predicted color of each sampled point can be computed as a weighted average of the image colors from the reference views, with $w$ serving as the weighting factor. Finally, the volume rendering is employed to derive predictions for the target pixel by blending all sampled points along the corresponding ray:

$$O(r) = \int_{t_n}^{t_f} T(t)\sigma(x(t))o(x(t))\,dt, \quad (7)$$

where $o \in \{s, c\}$ represents the output type, with $s$ denoting the semantic label and $c$ denoting color. Here, $O(r)$ is the final prediction for ray $r$, and $o(x(t))$ is the prediction for each sampled point. The accumulated transmittance $T(t)$ from the near plane $t_n$ to a point $t$ along the ray is given by:

$$T(t) = \exp\left(-\int_{t_n}^{t} \sigma(x(s))\,ds\right). \quad (8)$$

Here, $[t_n, t_f]$ defines the bounds of integration for the ray segment.

**Optimization.** Our GS²-GNeSF is optimized through a composite loss function, which integrates contributions from RGB, depth, semantic, and boundary losses. The RGB loss employs the mean squared error to ensure accurate color reproduction, by quantifying the discrepancies between predicted and actual colors. Depth accuracy is supervised using a smooth L1 loss that targets both predicted depth maps $\{\mathcal{D}^l\}_{l=1}^{L}$ and the rendered depth. For semantic segmentation, the loss $\mathcal{L}_{sem}$ applies cross-entropy to guide both the final semantic prediction and the predictions of the global context compensation's semantic branch, thus encapsulating a holistic understanding of scene semantics. Furthermore, to counteract the imbalance in boundary detection, a weighted binary cross-entropy loss [35] $\mathcal{L}_{bon}$ is utilized, which is commonly used to improve the model's boundary delineation capabilities.

The overall loss function integrates these individual components, weighted appropriately to balance their contributions:

$$\mathcal{L} = \lambda_{rgb}\mathcal{L}_{rgb} + \lambda_{depth}\mathcal{L}_{depth} + \lambda_{sem}\mathcal{L}_{sem} + \lambda_{bon}\mathcal{L}_{bon}. \quad (9)$$

Empirically, we set the parameters for the training loss of GS²-GNeSF as $\lambda_{rgb} = 0.75$, $\lambda_{depth} = 0.8$, $\lambda_{sem} = 0.25$, and $\lambda_{bon} = 5$.

## 4 EXPERIMENTS

### 4.1 Experiment Setup

**Datasets and Implementation Details.** To comprehensively assess our proposed method's effectiveness, we conduct experiments utilizing both synthetic and real datasets. Synthetic data experiments are carried out using the Replica dataset [28]. From this dataset, 12 scenes are selected for training, with the remaining unseen scenes designated for testing. For real-world data, we utilize the ScanNet dataset [8], a large labeled RGB-D dataset encompassing 2.5M views across 1513 scenes, annotated with 3D camera poses, surface reconstructions, and semantic segmentation. We train GS²-GNeSF on training splits commonly used in previous works [4, 23, 29], using novel scenes from validation splits to evaluate our model's generalization capabilities.

Following GNeSF [4], we pretrain Mask2Former as our feature extractor on the corresponding training set when training on Scan-Net [8] or Replica [28]. After that, our GS²-GNeSF is trained end-to-end on a single A40 GPU for 500k steps, employing the Adam

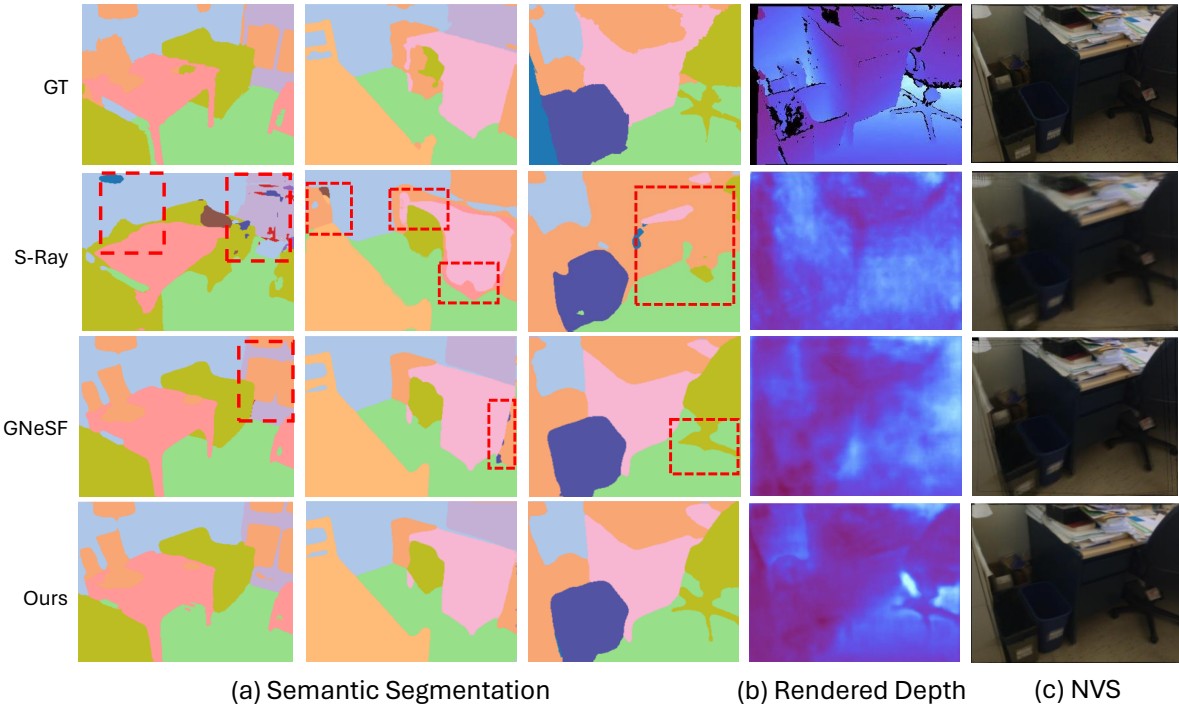

(a) Semantic Segmentation  (b) Rendered Depth  (c) NVS

**Figure 6: Qualitative comparison with SOTA methods on ScanNet [8] in generalization setting.** (a) The first three columns display the semantic segmentation results, where both S-Ray and GNeSF exhibit poor segmentation performance at object edges and interiors (indicated by the red dashed boxes). (b) The fourth column presents the rendered depth map, showing that neither S-Ray nor GNeSF successfully reconstructs the scene's accurate geometric structure. (c) The final column illustrates the Novel View Synthesis (NVS), with S-Ray showing noticeable artifacts and GNeSF appearing relatively blurry.

optimizer with an initial learning rate of $3e-4$, decaying to $1e-5$. The batch size of rays is set to 1024. In all our experiments, we sample 8 points for each ray, select $V = 8$ images as reference views, and render outputs at a resolution of $320 \times 240$.

**Evaluation Metrics.** Following previous works [4, 20], our evaluation framework employ mean Intersection-over-Union (mIoU), average accuracy (mAcc), and total accuracy metrics for segmentation quality assessment. Additionally, to evaluate the quality of novel view synthesis, we adopt peak signal-to-noise ratio (PSNR), and the structural similarity index measure (SSIM), following the established practices in the field.

In the experiments, to align with previous works, we evaluate our method under two settings: 1) assessing the *generalizability* of our pretrained model on unseen scenes without any fine-tuning, and 2) *fine-tuning* the model on unseen scenes before evaluating its performance on novel views.

## 4.2 Comparison with Baselines

To evaluate the effectiveness of our proposed GS²-GNeSF, we compare our model with three existing generalizable NeSF methods: S-Ray [20], GSNeRF [7], and GNeSF [4], under both generalization and finetuning setting. We assess the models through both quantitative and qualitative comparison, focusing on novel view synthesis and semantic segmentation. The results are detailed in Table 1 for quantitative analysis and Figure 6 for qualitative evaluation.

| No. | RGPG | GCC | GSI | PSNR | SSIM | mIoU | Total Acc | mAcc |
|---|---|---|---|---|---|---|---|---|
| 1 | × | × | × | 29.70 | 0.923 | 71.01 | 88.33 | 82.21 |
| 2 | ✓ | × | × | 30.33 | 0.921 | 72.12 | 89.14 | 82.64 |
| 3 | ✓ | ✓ | × | 31.34 | 0.912 | 73.40 | 88.92 | 83.23 |
| 4 | ✓ | × | ✓ | 31.62 | **0.937** | 73.21 | 89.13 | 83.33 |
| 5 | ✓ | ✓ | ✓ | **32.40** | 0.935 | **74.97** | **90.40** | **83.65** |

**Table 2:** Ablation Studies of our design choices on ScanNet [8].

As shown in Table 1, our model exhibits significant improvements in semantic segmentation (mIoU, Total Acc, and mAcc) across both synthetic and real datasets compared to baseline models. Additionally, the quality of our model's novel view images (PSNR and SSIM) also surpasses that of the baseline models. These significant improvements across all evaluation metrics indicate that our approach promotes a symbiotic relationship between reconstruction and semantic segmentation tasks, resulting in mutual reinforcement.

As depicted in Figure 6 (a), while S-Ray and GNeSF struggle with poor predictions at object boundaries or within objects, our model maintains superior segmentation quality in these challenging areas. The geometric representation offered by RGPG provides robust geometric information, evident from the improved depth map results showcased in Figure 6 (b). The quality of the reconstruction results in Figure 6 (c) further reveals that our novel view is crisper and exhibits fewer artifacts than S-Ray and GNeSF.

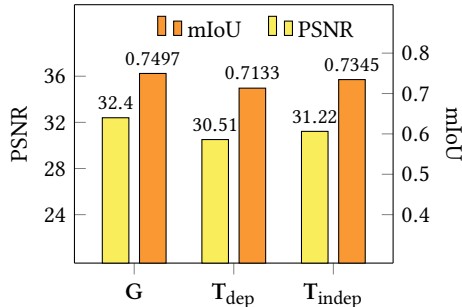

**Figure 7:** Ablation study of $T_{indep}$ and $T_{dep}$ in Sec. 3.1.3 on ScanNet.

## 4.3 Ablation Studies and Component Analysis

To validate the effectiveness of each component in our GS$^2$-GNeSF model, we systematically remove individual modules[3] from our model and present the results of ablation studies in Table 2.

**Robust Geometric Prior Generation (RGPG).** As shown in Table 2 (Row 1 & 2), the RGPG enhances performance in both image synthesis and semantic segmentation. This indicates the efficacy of RGPG in providing robust geometric features, as well as the impact of geometric-aware sampling in constructing semantic features.

Furthermore, we conduct an ablation study to analyze the impact of view-dependent features $T_{dep}$ and view-independent features $T_{indep}$ on constructing geometric features. The results, depicted in Figure 7, show that combining both (G) outperforms using only $T_{dep}$ or $T_{indep}$. Interestingly, $T_{indep}$ exhibits superior performance in both novel view synthesis and semantic segmentation, likely due to the robust geometric priors it introduces. Additionally, we observe that $T_{dep}$ effectively complements $T_{indep}$, enabling the model G to achieve the best performance in both tasks.

**Global Context Compensation (GCC).** It is noteworthy that when GCC is removed (*c.f.* Rows 1-3 in Table 2), we adopt a soft voting mechanism for label prediction, following the approach outlined in GNeSF [4], which yields better results compared to direct inference from features [7, 20]. As shown in Table 2, even against such as strong baseline (Row 2), the addition of the GCC module (Row 3) can still effectively enhance the model's semantic segmentation quality (mIoU). Additionally, we observe that GCC contributes to improvements in both novel view synthesis quality (PSNR) and semantic segmentation result (mIoU), as evidenced by comparisons between Rows 2 and 3, and Rows 4 and 5.

To further evaluate the individual and combined contributions of the semantic and boundary branches in GCC's dual-branch encoder, we conduct an ablation study with three setups: a) both branches active, b) only semantic, and c) only boundary operational. The results presented in the upper section of Table 3 indicate that the semantic branch contributes to the segmentation results, while the boundary branch notably enhances the rendering quality (PSNR & SSIM). Finally, the joint activation of both branches (Row 8) yields the best results, improving both segmentation and rendering, which can also be observed by the segmentation visualizations in Figure 8, particularly at object edges. Interestingly, when only having the

---

[3]When either the GCC or GSI module is employed, the RGPG is not removed. This is due to the GCC's dependence on the depth prior from the RGPG for building the global semantic context, and the GSI's use of geometric features derived from the RGPG's cost volume as inputs.

| No. | Description | PSNR | SSIM | mIoU | Total Acc | mAcc |
|---|---|---|---|---|---|---|
| 1 | w/o GCC | 31.62 | 0.937 | 73.21 | 89.13 | 83.33 |
| 2 | only semantic branch | 31.22 | 0.944 | 74.26 | 89.24 | 83.55 |
| 3 | only boundary branch | 32.35 | **0.962** | 73.81 | 88.78 | 83.02 |
| 4 | w/o GSI | 31.34 | 0.912 | 73.40 | 88.92 | 83.23 |
| 5 | only inter-view phase | 32.10 | 0.912 | 73.86 | 89.51 | 83.51 |
| 6 | only intra-view phase | 31.58 | 0.922 | 73.55 | 89.30 | 83.52 |
| 7 | intra-view → inter-view | 32.37 | 0.941 | 74.49 | 90.19 | 83.32 |
| 8 | GS$^2$-GNeSF | **32.40** | 0.935 | **74.97** | **90.40** | **83.65** |

**Table 3:** Ablation Studies of GCC and GSI on ScanNet.

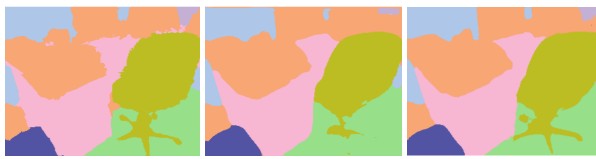

GT        Ours w/o BD        Ours with BD

**Figure 8:** Impact of Boundary Detection (BD) on semantic segmentation.

boundary branch, the model achieves the best performance on the SSIM metric, reaching the second highest value on PSNR among all results in Table 3. This suggests that the primary contribution of the GCC to novel view synthesis lies in the boundary-aware features, highlighting the significance of our geometry-semantics synergy in the GCC module.

**GeoSem Interaction (GSI).** By comparing Rows 2 and 4, and Rows 3 and 5 in Table 2, we can observe that the GSI facilitates simultaneous enhancements in both semantic segmentation and novel view synthesis tasks. This mutual enhancement demonstrates the effective integration of geometric and semantic features by the GSI. With the assistance of GSI, geometric and semantic features are able to acquire mutually beneficial information from each other, ultimately improving the quality of both novel view synthesis and semantic segmentation.

Additionally, to validate the design choice of our dual-directional interactive attention in the GSI, we conduct an ablation study with three alternatives: a) solely inter-view attention, b) solely intra-view attention, and c) reversed phase order: applying intra-view attention first, followed by inter-view attention. The results in the lower section of Table 3 demonstrate that individual attention phases yield slight improvements in rendering and segmentation, while their combined use maximizes performance. The sequence of phases has minimal impact on the outcome, confirming the robustness of their integration.

## 5 CONCLUSION

In this paper, we propose a novel approach, GS$^2$-GNeSF, that significantly advances the integration of geometric and semantic understanding in 3D scene perception. By developing innovative components such as the robust geometric prior generation, global context compensation, and geosem interaction, our method not only enhances the visual rendering capabilities but also significantly improves the performance of semantic segmentation. Our experimental results, conducted on both synthetic and real datasets, confirm the superior performance of our approach over sota methods.

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
