# OpenReview forum: "GS$^2$-GNeSF: Geometry-Semantics Synergy for Generalizable Neural Semantic Fields"
_acmmm.org/ACMMM/2024/Conference — MM2024 Poster_

### Official Review · Reviewer_sWqy · 2024-05-04

**Rating:** 4
**Confidence:** 4

**Summary:**

This paper introduces a robust geometric prior generator to generate the cost volumes and depth prior, which aid in constructing geometric features and facilitating geometric-aware sampling. With the depth priors, it additionally constructs a global semantic context for the target view. In conclusion, this paper is novel enough and achieves remarkable results. I’m willing to raise the score if the authors can solve my concerns in rebuttal.

**Strengths:**

1. This paper is well written and easy to read.
2. The proposed methods (I.e. RGPG, GCC and GSI) is reasonable and make valuable contributions for the final performance.
3. The usage of Mask2Former is novel in Generalized NeRF.

**Limitations:**

1. There has been work to explore the combination of NeRF and semantic network, which should be introduced and compared: H Li, et al. GP-NeRF [CVPR 2024]
2. Since Generalized NeRFs like IBRNet, GPNR, and GNT render few pixels in a single batch, they are unable to conduct semantic interactive attention. Is the proposed RGPG rendering the whole image in a single batch? If so, what’s the FPS? If not, how does the GS^2-NeSF overcome this problem?
3. Since this paper use Mask2Former to generate pseudo labels and semantic features for supervision and achieve remarkable performance improvement. I’m wondering if using pre-train Mask2Former provides more accurate segmentation masks than the provided GT, because the original masks in ScanNet is pretty rough.
4. As illustrated  in Fig. 3, the Semantic Feature Construction only use 8 sampled points in each sampled ray, while Global Semantic Context use a projection-reprojection strategy. Is the GSC plays the bigger role / more contributions for the performance?
5. Although it shows remarkable segment and reconstruction results, the reviewer hopes the author to demonstrate more visualization results. Moreover, depth priors play an important role in the paper. However, in Figure 6, it seems the depth visualization is not good enough? Please explain it.

**Suitability:**

3

---

### Official Review · Reviewer_iimt · 2024-05-20

**Rating:** 4
**Confidence:** 2

**Summary:**

This paper argues that previous generalizable NeSF fail to fully exploit the geometric and semantic features as well as their mutual interactions, resulting in suboptimal performance in both novel-view image synthesis and semantic segmentation. To address this limitation, the authors propose the RGPG, the GCC, and the GSI in the GS2-GNeSF framework. The reviewer observes substantial improvement compared with previous SOTA methods. Besides, the ablation study shows the contributions of the proposed modules.

**Strengths:**

+ Although this paper seems to be an improvemental work based on GNeSF, the contributions of this work cannot be ignored. Adding the depth prior makes senses and the dual-directional interactive attention seems to be better than the voting operation in GNeSF.
+ The performance improvement made by this work is large and convincing.

**Limitations:**

- More discussion about the main idea of this paper should be included. The reviewer wants to see why previous methods cannot fully exploit the geometric and semantic features as well as their mutual interactions, and by doing what, this work achieve such a good performance.
- More visualized results of novel views should be shown. The reviewer finds only few novel views of limited scenes in the main paper and supplemental materials.
- Some experimental details are not clear: what is the framework like without the RGPG?

**Suitability:**

3

---

### Official Review · Reviewer_Zr6F · 2024-05-23

**Rating:** 5
**Confidence:** 3

**Summary:**

By deeply integrating geometric and semantic features, this paper addresses the problem of insufficient utilization of geometric and semantic features and their interaction in the existing methods. Specifically, using a robust geometric prior generator (RPRG), global context compensation (GCC), and geometric-semantic interaction (GSI), semantic segmentation and new view synthesis have achieved results beyond the existing models.

**Strengths:**

1. **A new method is proposed to solve the problems of novel view synthesis with semantic segmentation**. The paper introduces three modules on the basic NeSF, which improves the accuracy of semantic segmentation and the quality of generated images.

2. **Detailed and helpful figures are provided**. The figures depicted in this paper make it clearer and easy to understand. The results of the paper are also obvious in comparison with the results of other models.

3. **The intuition and motivation are great**. The motivation is insightful, (i.e., geometric understanding (e.g. volume density) and semantic understanding (e.g. semantic labels) carry mutually beneficial information).

4. **The comparative experimental data are compelling**. The results are compared with the baseline from both qualitative and quantitative aspects and show the effectiveness of the framework.

**Limitations:**

1. **Lack of more contrast for view composition**. Note that the view synthesis feature only uses one generated image for comparison against the baseline, which is quite rare given the variety of 3D views. Not only can you add composite photos of the same object viewed from different directions for comparison, but you can also add new view photos of several different objects.

2. **The part about global context compensation (GCC) in the ablation experiment**. It is noted that the ablation experiment data show that deleting GCC generates a higher structural similarity index (SSIM) score for new views while adding only the semantic branch of GCC or adding only the boundary branch of GCC results in higher results than deleting GCC and retaining GCC. Does this mean that the two modules work together to produce negative effects? In addition, in the experiment of adding only semantic branches, the PSNR of generating new attempts is reduced, and the SSIM is hardly improved. Does this mean that the introduction of semantic branch modules has little effect on the overall model?

3. **Limited contribution.** The GeoSem Interaction includes the Inter-view phase and Intra-view phase, which is quite similar to the design in Semantic Ray. And it seems like only incorporating depth information into the architecture as NeRF representation is not geometric-aware. What about using Gaussian Splatting as an explicit 3D representation that inherently contains rich geometric information?

**Suitability:**

2

---

### Official Review · Reviewer_LT8c · 2024-05-24

**Rating:** 3
**Confidence:** 3

**Summary:**

In this paper, the authors propose a new NERF approach that not only enhances the visual rendering capability but also significantly improves the performance of semantic segmentation by developing innovative components such as robust geometric a prior generation, global context compensation and geosem interaction. It has good robustness and adaptivity in unknown scenarios. The proposed framework is novel and experimentally detailed, but there are still a series of problems that limit the application of the proposed approach.

**Strengths:**

The proposed NERF approach improves the performance of generalized neural semantic fields through the integrated construction and synergistic interaction of geometric and semantic features. An effective two-way interaction attention mechanism is proposed to facilitate the deep interaction between the enhanced geometric and semantic features. Experiments conducted on both synthetic and real datasets show that the proposed nerf method outperforms existing methods in both new view and semantic map synthesis, highlighting its effectiveness in generalizing the neural semantic field in unknown scenarios. Moreover, the paper is technically fine and clear, and the results on the different datasets compare very favorably with the different baselines. The algorithm is simple and training seems easy.

**Limitations:**

1: The author does not provide a detailed description of the semantic feature extraction process. It is well known that the extraction of semantic features is usually a kind of lossy compression, will the Reconstructed semantic information negatively affect the NeRF generation results? In addition, does the fusion of semantic features and geometric information lead to the lack of positional information?

2. In the autopilot and SLAM domains, there are strict requirements on rendering time and resource consumption, but this paper lacks a comparative analysis of generation time and memory consumption. In addition, this paper does not consider the semantic information of the time dimension and whether the generated results will generate artifacts when rendering object motion.

3. The description of the experimental simulation section is not comprehensive enough, it is advisable to add some experimental results that compare with SOTA methods (Generative NeRF and 3D Gaussian Splatting ).

4: It is better to describe the current research status about the semantic NeRF more systematically. Moreover, the figures in your paper are a bit blurry.

**Suitability:**

2

---

### Meta-Review · Area_Chair_zssH · 2024-06-27

**Recommendation:** Accept (Poster)
**Confidence:** 4

**Metareview:**

The paper received mixed scores of acceptance and borderline. As the problem it solves is quite new and interesting (learning NeRF with semantics), I recommend a decision of acceptance. The authors should carefully address the concerns of the reviewers in the final version, such as the time and memory usage.